# Gene polymorphisms and serum levels of BDNF and CRH in vitiligo patients

**Assiya Kussainova[1], Laura Kassym[2]\*, Nazira Bekenova[1], Almira Akhmetova[1], Natalya Glushkova[3], Almas Kussainov[4], Zhanar Urazalina[5], Oxana Yurkovskaya[6], Yerbol Smail[7], Laura Pak[8], Yuliya Semenova[9]**

1 Department of Dermatovenerology and Cosmetology, NJSC "Semey Medical University", Semey, Republic of Kazakhstan, 2 School of Medicine, Nazarbayev University, Nur-Sultan, Republic of Kazakhstan, 3 Department of Epidemiology, Biostatistics & Evidence Based Medicine, Al-Farabi Kazakh National University, Almaty, Republic of Kazakhstan, 4 Department of Psychiatry and Narcology, NJSC "Astana Medical University", Nur-Sultan, Republic of Kazakhstan, 5 Department of Emergency Medicine, NJSC "Semey Medical University", Semey, Republic of Kazakhstan, 6 Department of Personalized Medicine, NJSC "Semey Medical University", Semey, Republic of Kazakhstan, 7 Department of Infectious Diseases and Immunology, NJSC "Semey Medical University", Semey, Republic of Kazakhstan, 8 Department of Clinical Oncology and Nuclear Medicine, NJSC "Semey Medical University", Semey, Republic of Kazakhstan, 9 Department of Neurology, Ophthalmology, Otorhinolaryngology, NJSC "Semey Medical University", Semey, Republic of Kazakhstan

\* laura.kassym@gmail.com

**Data Availability Statement:** All relevant data are within the article and its Supporting Information files.

## Abstract

### Background

Vitiligo is one of the most common hypomelanoses, in which the destruction of functioning melanocytes causes depigmentation of the skin, hair and mucous membranes. The genes encrypting brain-derived neurotrophic factor (BDNF) and corticotropin releasing hormone (CRH) might be the conceivable contributors to the development of vitiligo. This study was aimed at investigation of the serum levels of BDNF and CRH as well as their selected single nucleotide polymorphisms (SNPs) in vitiligo patients in comparison with the healthy controls.

### Methods

The cross-sectional study was carried out between October 2020 and June 2021 in 93 vitiligo patients (age range from 23 to 48 years) and 132 healthy controls (age range from 24 to 52 years). The psychological status of study participants was evaluated using the Generalized Anxiety Disorder-7 (GAD-7) scale. Serum levels of BDNF and CRH were measured with the help of a commercially available sandwich enzyme-linked immunosorbent assay (ELISA) kit. Genotyping for the rs11030094 polymorphism of the *BDNF* gene and for the rs242924 polymorphism of the corticotropin releasing hormone receptor 1 *(CRH-R1)* gene was performed by a real-time polymerase chain reaction (PCR).

### Results

There was a significant relationship between the *CRH-R1* rs242924 and *BDNF* rs11030094 polymorphisms and vitiligo. Moreover, serum levels of neurotransmitters differed

**Funding:** The author(s) received no specific funding for this work.

**Competing interests:** The authors have declared that no competing interests exist.

significantly between vitiligo and control groups and were associated with the *CRH-R1* rs242924 and *BDNF* rs11030094 SNPs.

## Conclusions

Our findings demonstrated the association between *CRH-R1* rs242924 and *BDNF* rs11030094 polymorphisms and vitiligo. Further studies need to be carried out in vitiligo patients to confirm the results observed.

## 1 Background

Vitiligo is an acquired pigmentary disorder of the skin and mucous membranes, which manifests as white macules that appear due to a selective loss of melanocytes [1]. According to major population surveys, the estimated global prevalence of vitiligo is 0.5–2%, which can reach 8.8% in some areas of Asia [2]. The recent evidence demonstrated the complex patterns of intrinsic and adaptive immune mechanisms, oxidative stress, and disturbance of neuronal maintenance in skin [3]. Clinically, the non-symptomatic whitish macules on skin in vitiligo patients are not as onerous as the signs of other chronic cutaneous diseases. Nevertheless, vitiligo may serve as a cause for the wide range of psychological impairments resulting in violation of mental well-being and quality of life [4–6]. To this end, the better understanding of pathophysiology of vitiligo might be achieved by investigation of the interplay between genetic, neurohumoral, and environmental factors in the onset and progression of the disease [7–10].

Brain-derived neurotrophic factor (BDNF) is the growth factor that regulates survival, differentiation, and synaptic plasticity of neurons in the central neural system as well as the peripheral nerve fibers [11]. The data on the contribution of BDNF levels to pathophysiology of different cutaneous disorders remain controversial. Some studies reported the increased BDNF levels in patients with atopic dermatitis and androgenic alopecia [12, 13]. On the contrast, by Yanik et al. [14], the serum levels of BDNF were found to be reduced in vitiligo patients. Keeping in mind the contribution of neuropsychiatric abnormalities to vitiligo development, the same results of BDNF measurements in population with mental problems are expected. The lower BDNF levels were detected in patients with stress-induced depressive and anxious behaviors [15].

Corticotropin-releasing hormone (CRH) regulates the hypothalamic-pituitary–adrenal axis (HPA) through activation of its receptor (corticotropin-releasing hormone receptor 1 (CRH-R1)). The CRH gene is also expressed in extracranial tissues, including human skin, and can be synthesized by a variety of immune cells. Human mast cells may contain sufficient amounts of CRH and other stress peptides to mediate its autoimmune and anti-inflammatory effects [16]. Expression of CRH and CRH-R1 mRNA also has been detected in cultured human melanocytes and melanoma cells [17]. Research on vitiligo demonstrated that expression of the CRH-R1 gene is increased in both vitiligo unaffected and vitiligo affected skin. Furthermore, the increased expression of CRH and CRH-R1 correlated with high scores obtained on the Social adaptation assessment scale, which can be explained by the fact that hyperactivation of the cortical-releasing system in skin increases neuroendocrine sensitivity to stress [18].

Assigning a pivotal role of the abovementioned signaling substances in skin disorders, we estimated the genes encrypting *BDNF* and *CRH* as the conceivable contributors to the development of vitiligo. Current data show that some *BDNF* single nucleotide polymorphisms (SNPs) are associated with neurological, psychological, and cognitive impairments [19–21]. Recently,

Hennings et al. [22] reported that *BDNF* rs11030094 polymorphism was associated with anti-depressant treatment response in patients with different mental disorders. *CRH* gene disturbances are also recognized as the possible risk factors for dysregulation of stress response. For instance, the *CRH-R1* rs242924 polymorphism was associated with the higher incidence of panic disorders [23].

To the best of our knowledge, no previous study examined the possible role of either *BDNF* rs11030094 polymorphism or *CRH-R1* rs242924 polymorphism as contributing factors to vitiligo. Thus, we aimed to investigate the serum levels of BDNF and CRH as well as their selected SNPs in vitiligo patients and healthy controls.

## 2 Materials and methods

### 2.1 Study population

This was a cross-sectional study of people with vitiligo and age- and sex-matched healthy comparators, which was based in the dermatology department of General Hospital # 2 in Semey City, Kazakhstan from October 1, 2020, to June 30, 2021. All vitiligo patients included in the study had a clear diagnosis of vitiligo established by their physicians and were identified via the electronic medical records. The percentage of skin lesions was calculated using the Vitiligo Extent Score (VES) online calculator [24]. The psychological status of study participants was identified using Generalized Anxiety Disorder-7 (GAD-7) scale. The GAD-7 scores range from 0 to 21, with 0–4 none, 5–9 mild, 10–14 moderate and 15 and more severe levels of anxiety symptoms [25]. The non-vitiligo (control) participants were matched by age and sex as the comparator group, and were recruited through community advertisement. All participants were enrolled in the study after giving a written informed consent. For the patient to be considered eligible for the study, the following inclusion criteria had to be met: (i) age of 16 years and older, (ii) a verified diagnosis of vitiligo, and (iii) an ability to read and understand Russian or Kazakh. Patients were ineligible for this investigation if they: (i) were physically or mentally unable to give informed consent, (ii) had a history of any psychiatric disorder, (iii) had some somatic conditions, such as heart, pulmonary diseases or diabetes mellitus that affected their mental status, (iv) had other skin problems (acne, psoriasis, eczema, etc.)., and (v) presented with autoimmune or inflammatory diseases.

The study was approved by the Ethics Committee of Semey Medical University (approval # 2; dated October 18, 2019) within guidelines established by the 1964 Declaration of Helsinki.

### 2.2 Blood sampling, serum BDNF and CRH level analysis

Blood sampling was performed during the daytime from 8 am, in coordination with the participants' visits to the hospital. Five milliliters of venous blood were collected into ethylenediaminetetraacetic acid (EDTA) tube and centrifuged during 30 minutes after collection. Centrifugation was carried out at 3000 rpm for 15 minutes. Plasma and buffy coat were extracted and stored at -80˚C until analysis was carried-out at the Scientific-Research Laboratory Center of Semey Medical University. Serum BDNF levels were determined with the help of a commercially available sandwich enzyme-linked immunosorbent assay (ELISA) kit for human BDNF (Cloud-Clone Corp., USA). The test was performed in accordance with the manufacturer's instructions. Plasma BDNF levels were calculated based on a standard curve generated by serially diluted BDNF standards. An intra-assay coefficient of variance (CV) of less than 10% was considered acceptable, and the mean intra-assay CV obtained was 4.9%. The levels of CRH in the serum were also detected by an ELISA kit (Cloud-Clone Corp., USA). The test methodology followed the manufacturer's protocol.

### 2.3 DNA extraction, BDNF and CRH-R1 genotyping

DNA extraction from blood samples (100 μl) was carried out with the ready-made commercial GeneJET Mini kit (Thermo Scientific, Vilnius, Lithuania) in accordance with the manufacturer's instructions. DNA concentration was assessed using fluorometer Qubit 4 (Thermo Scientific, Walthem, MA, USA), using Invitrogen reagent kits (Thermo Fisher Scientific, Eugene, OR, USA). The isolated DNA was frozen and stored at -20 C degrees. Genotyping for the rs11030094 polymorphism of the *BDNF* gene and rs242924 polymorphism of the *CRH-R1* gene was performed by a real-time PCR on a QuantStudio 5 (Applied biosystems, Thermo Fisher Scientific) device using ready-made mixtures of primers and TaqMan probes in the presence of the TaqMan Genotyping Master mix reagent (all reagents manufactured by Life Technologies, Foster City, CA, USA) and 10 ng of DNA in a total volume of 5 μl. The amplification program included pre-denaturation at 60 C degrees for 30 seconds, then at 95 C degrees for 10 minutes and 40 cycles at 95 C degrees for 15 seconds and finally, at 60 C degrees for 60 seconds for all SNPs.

### 2.4 Statistical analysis

Statistical evaluation to identify associations between polymorphisms of the *CRH-R1* and *BDNF* genes and presence of vitiligo was carried out via Gene Expert program (http://gen-exp.ru/calculator_or.php) using the chi-squared test (χ2) and the odds ratio (OR) with 95% confidence intervals (CI). The ratio of frequencies of genotypes and alleles was checked for compliance with the Hardy-Weinberg equilibrium (HWE).

The content of CRH and BDNF neurotransmitters in blood serum was analyzed using medians and quartiles (Me, Q1 and Q3, respectively). The levels of neurotransmitters in different genotype subgroups both in patients and controls were compared using Kruskal-Wallis test. The rates of CRH and BDNF within the same genotype in vitiligo patients and healthy controls were compared using the Mann–Whitney U test. All analyses were performed using the SPSS 20.0 package at a significance level of $p < 0.05$.

### 3 Results

The baseline characteristics of cases and controls are summarized in Table 1. This study consisted of 93 vitiligo patients (age range from 23 to 48 years) and 132 healthy controls (age range from 24 to 52 years). No difference was observed between two groups according to age, gender, and ethnicity. Nearly half of the participants had a university degree. More than half of the participants were married, and the same proportion had both parents alive. The skin types II and III prevailed among both vitiligo patients and healthy controls. The GAD-7 scores were significantly higher in cases as compared with the healthy controls (p<0.001). The serum levels of BDNF and CRH are presented in S1 Fig. The correlation of ELISA levels and VES was not significant (S1 Table).

### 3.1 rs242924 polymorphism

Both case and control groups were in HWE. Genotype and allele frequencies are summarized in Table 2. The T-allele and TT genotype were at risk of having vitiligo (Table 2).

### 3.2 Association of rs242924 with serum CRH levels

Serum CRH levels in individual genotypes of SNP rs242924 were evaluated between the case and control groups. It was found-out that CRH level was significantly higher in TT-genotype subgroup of cases as compared with the healthy controls: 7.30 (2.41–8.84) ng/mL vs. 2.97 (1.87–4.09) ng/mL (p = 0.001) (Table 3).

**Table 1. Demographic and clinical characteristics of the study participants.**

| | | Cases | | Controls | | | |
|---|---|---|---|---|---|---|---|
| | | N | % | N | % | Statistical test | p-value |
| Age (years), median and 25<sup>th</sup>-75<sup>th</sup> percentile | | 35 | 23–48 | 38 | 24–52 | 0.752* | 0.453 |
| Gender | Females | 53 | 57.0 | 76 | 57.6 | 0.008** | 0.930 |
| | Males | 40 | 43.0 | 56 | 42.4 | | |
| Ethnicity | Kazakh | 83 | 89.2 | 125 | 94.7 | 3.063** | 0.216 |
| | Russian | 9 | 9.7 | 7 | 5.3 | | |
| | Other | 1 | 1.1 | 0 | 0.0 | | |
| Education level | HS or less | 6 | 6.5 | 8 | 6.1 | 2.499** | 0.287 |
| | College or less | 41 | 44.1 | 45 | 34.1 | | |
| | HE or less | 46 | 49.5 | 79 | 59.8 | | |
| Marital status | Single | 28 | 30.1 | 45 | 34.1 | 1.060** | 0.787 |
| | Married | 53 | 57.0 | 67 | 50.8 | | |
| | Divorced | 5 | 5.4 | 10 | 7.6 | | |
| | Widow | 7 | 7.5 | 10 | 7.6 | | |
| Having parents | Both parents are alive | 50 | 53.8 | 81 | 61.4 | 1.572** | 0.456 |
| | There is only one alive parent | 23 | 24.7 | 30 | 22.7 | | |
| | No parents | 20 | 21.5 | 21 | 15.9 | | |
| Fitzpatrick skin type | 1 | 9 | 9.7 | 16 | 12.1 | 1.392** | 0.707 |
| | 2 | 44 | 47.3 | 65 | 49.2 | | |
| | 3 | 29 | 31.2 | 41 | 31.1 | | |
| | 4 | 11 | 11.8 | 10 | 7.6 | | |
| GAD-7 groups | None | 50 | 53.80 | 109 | 82.60 | 27.639** | <0.001 |
| | Mild | 30 | 32.30 | 22 | 16.70 | | |
| | Moderate | 8 | 8.60 | 1 | 0.80 | | |
| | Severe | 5 | 5.40 | 0 | 0.00 | | |

*–test of difference was Mann Whitney U-test

**–test of difference was Chi-square test

HS–Higher School; HE–Higher Education

**Table 2. Genotype distribution and allele frequencies of *CRH-R1* rs242924.**

| Alleles | Cases N = 93 (%) | Controls N = 132 (%) | P | OR (95% CI) |
|---|---|---|---|---|
| G | 24 (25.3%) | 52 (39.4%) | 0.002 | 0.52 (0.34–0.79) |
| T | 69 (74.7%) | 80 (60.6%) | | 1.92 (1.27–2.90) |
| Genotypes | | | | |
| GG | 6 (6.5%) | 28 (21.2%) | 0.007 | 0.26 (0.10–0.65) |
| GT | 35 (37.6%) | 48 (36.4%) | | 1.06 (0.61–1.83) |
| TT | 52 (55.9%) | 56 (42.4%) | | 1.72 (1.01–2.94) |

P < 0.05 is significant

OR–Odds Ratio (95% CI—95% Confidence Interval)

A multiplicative model (df = 1) was used to identify the association of alleles with the disease

A general inheritance model (df = 2) was used to identify the association of genotypes with the disease

**Table 3. Statistical association between serum CRH levels and *CRH-R1* rs242924 genotypes in the case and control groups.**

| Study group | GG | GT | TT | p* |
|---|---|---|---|---|
| Cases N = 93 | 5.91 (2.66–7.36) | 5.91 (2.48–7.71) | 7.30 (2.41–8.84) | 0.37 |
| Controls N = 132 | 3.19 (2.40–3.84) | 3.47 (2.91–3.95) | 2.97 (1.87–4.09) | 0.43 |
| p** | 0.09 | 0.08 | 0.001 | |

Data are given as Me (Q$_1$-Q$_3$)

p*–P value of CRH levels between GG, GT, and TT genotypes in both case and control groups

p**–P value between of CRH levels within genotype subgroup

### 3.3 rs11030094 polymorphism

Genotype distribution was in HWE for both case and control groups. Table 3 demonstrates the distribution of alleles and genotypes in vitiligo patients and healthy controls. The G-allele was significantly associated with the risk of vitiligo, whereas AA-genotype was found-out to be the factor decreasing the risk of the disease (Table 4).

### 3.4 Association of rs11030094 with serum BDNF levels

There was no significant difference in serum BDNF levels among the genotypes of the case and control groups (Table 5). Healthy controls in both GG-genotype and GA-genotype subgroups had significantly lower serum BDNF levels as compared with the vitiligo patients.

## Discussion

The aim of our study was to investigate the serum levels of BDNF and CRH, as well as their individual SNPs, in patients with vitiligo and healthy comparators. After analyzing the laboratory data of the two groups, we were able to establish a significant relationship between the *CRH-R1* rs242924 and *BDNF* rs11030094 polymorphisms and vitiligo. Moreover, serum levels of neurotransmitters differed significantly between vitiligo and control groups and were associated with the *CRH-R1* rs242924 and *BDNF* rs11030094 SNPs. These findings may point to a potential role played by the aforementioned genetic variations in the development of vitiligo.

There is a wide spectrum of studies related to the investigation of *CRH-R1* gene rs242924 polymorphism. Most of them focus on the abovementioned SNPs to elucidate their contribution to the pathogenesis of neuropsychological and behavioral impairments in different

**Table 4. Genotype distribution and allele frequencies of *BDNF* rs11030094.**

| Alleles | Case | Control | | OR (95% CI) |
|---|---|---|---|---|
| G | 53 (57.5%) | 61 (46.2%) | 0.002 | 1.58 (1.08–2.30) |
| A | 40 (42.5%) | 71 (53.8%) | | 0.63 (0.43–0.93) |
| Genotypes | | | | |
| GG | 28 (30.1%) | 33 (25.0%) | 0.01 | 1.29 (0.71–2.34) |
| GA | 51 (54.8%) | 56 (42.4%) | | 1.65(0.97–2.81) |
| AA | 14 (15.1%) | 43 (32.6%) | | 0.37(0.19–0.72) |

P < 0.05 is significant

OR–Odds Ratio (95% CI—95% Confidence Interval)

A multiplicative model (df = 1) was used to identify the association of alleles with the disease

A general inheritance model (df = 2) was used to identify the association of genotypes with the disease

**Table 5. Statistical association between serum BDNF levels and *BDNF* rs11030094 genotypes in the case and control groups.**

| Study group | GG | GA | AA | p$^*$ |
|---|---|---|---|---|
| Case (n = 93) | 2.52 (2.26–3.23) | 2.62 (1.83–3.0) | 2.71 (1.72–3.0) | 0.77 |
| Control (n = 132) | 3.31 (2.55–4.27) | 3.14 (2.17–3.89) | 2.91 (2.22–4.08) | 0.72 |
| p$^{**}$ | 0.032 | 0.007 | 0.15 | |

Data are given as Me (Q$_1$-Q$_3$)

p$^*$–P value of CRH levels between GG, GT, and TT genotypes in both case and control groups

p$^{**}$–P value between of CRH levels within genotype subgroup

populations [26–28]. Some studies are mainly concerned with the polymorphism of the *CRH-R1* rs242924 gene in autoimmune disease. Such, Sato et al. [29] found that T-alleles rs7209436 and rs242924 are associated with the increased prevalence of diarrhea symptoms in patients with autoimmune irritable bowel syndrome. Several studies also examined cutaneous expression of CRH and CRH-R1 by performing a skin biopsy. For instance, Cemil et al. [30] found a statistically significant increase in CRH-R1 expression in psoriatic lesions and hypothesized that CRH-R1 may play a role in the pathogenesis of psoriasis. In addition, Papadopoulou et al. [31] pointed out that chronic urticaria expresses higher levels of CRH-R1 and mast cells as compared with the normal foreskin, breast skin and cultured human keratinocytes. These results implicate the role of CRH-R1 in chronic urticaria, which is often exacerbated by stress. In the literature, there is only one report describing the role of CRH and CRH-R1 in vitiligo. The authors found a substantial rise in expression of CRH and CRH-R1 in both damaged and undamaged skin of patients with vitiligo and observed a significant direct association with stress levels [18]. To our knowledge, the present study is the first that reports on association between vitiligo and SNPs of the *CRH-R1* gene. Our finding revealed that T-allele and TT-genotype were significantly associated with vitiligo, and the level of CRH was significantly higher in cases with TT-genotype subgroup. Unfortunately, there is a lack of evidence reporting the effects of the polymorphisms studied on serum levels of the mediators. Still, according to Tsartsali et al. [32], two SNP interactions (rs242941 and rs1876828 of the *CRH-R1* gene) were linked to the serum cortisol levels.

It is hypothesized that BDNF is a neurotransmitter, which plays a crucial role in the up- and down regulation of different pathways controlling synaptic plasticity. Yanik et al. [14] were the first who described the reduced serum levels of BDNF in vitiligo patients as compared with the healthy controls. Our recent study also demonstrated the decreased level of serum BDNF and increased level of serum CRH in vitiligo patients in comparison with the healthy controls [33]. Unfortunately, we were unable to find studies related to the expression of *BDNF* rs11030094 polymorphism in skin disorders. However, there is a number of publications devoted to the investigation of *BDNF* rs6265 (Val66Met) gene polymorphism in non-infectious inflammatory skin diseases. For instance, Hoffjan et al. [34] considered that the variation in the *BDNF* rs6265 gene is unlikely to play a substantial role in the pathogenesis of atopic dermatitis (AD), even though BDNF serum levels were considered to be a useful indicator of disease activity in AD patients. Quan et al. [35] notified that the combined effects of *BDNF* Val66Met and higher body mass index (C25) increased the risk and clinical severity of psoriasis vulgaris in Chinese Han population.

Our data revealed the significantly lower levels of BDNF in cases from GG-genotype and GA-genotype subgroups. There is no data on associations of SNPs with serum levels of BDNF in patients with the skin disorders. Few studies described such associations between SNPs of

other mediators and their blood levels. For instance, a significant association between IL-6 572 gene polymorphism and its serum level was reported in Egyptian patients with keloid [36]. Another study showed that TT genotype of IL-19 might be a hereditary risk factor for acne vulgaris development, and it was associated with elevated IL-19 serum levels, which could be a marker of acne severity [37]. Therefore, the potential significance of *BDNF* SNPs in the pathogenesis of skin disorders is in the process of investigation. Nevertheless, some evidence on association of vitiligo with gene polymorphisms exists. Tumor necrosis factor (TNF)-α- 308 G/A gene polymorphism was the most investigated in vitiligo patients. The recent meta-analysis of Giri et al. [38] suggested the involvement of susceptible 'A' allele with vitiligo susceptibility in overall population, whereas the previous studies failed to reveal that TNF-α-308 G/A polymorphism is a genetic risk factor for vitiligo [39, 40].

Corticotropin-releasing hormone is one of the main neuropeptides that plays a crucial role in controlling the hypothalamic–pituitary–adrenal axis and the systemic answer to stress [41, 42]. CRH is mostly generated in the central nervous system. Nevertheless, CRH and its receptors are also expressed in numerous peripheral tissues, including skin [10]. Cutaneous CRH is assumed to control different skin functions essential for local homeostasis [43]. The effects of CRH are indirect and are conveyed through two main types of receptors called CRH-R1 and CRH-R2 [44]. In humans, CRH-R1 is mostly expressed in all major cellular populations, whereas CRH-R2 is expressed only in hair follicle keratinocytes, sebocytes, and dermal papilla fibroblasts [45, 46]. Inside the skin, CRH and its receptors are variably expressed in dependence with the cell type and presence/absence of inflammation or disease. CRH has direct immunomodulatory effects on different immune cells. CRH provokes various signaling pathways via CRH-R1 activation, which influences proliferation, differentiation, apoptosis and pro- or anti-inflammatory activities of skin cells [47]. CRH-caused activation of mast cells (via CRHR-1a) and subsequent vascular permeability significantly contribute to inflammation [48]. Such, Ziegler et al. [10] reported that numerous skin disorders like acne, AD, psoriasis, seborrhoic eczema, prurigo, urticaria, and alopecia areata can be provoked by stress, probably also being mediated by CRH. In addition, corticotropin-releasing hormone can also cause the release of melanin-destroying molecules and pro-apoptotic molecules, which makes the skin under stress vulnerable and leads to hypopigmentation, especially since melanocytes express the corticotropin-1 releasing hormone receptor [49]. Therefore, neuroimmune interactions are capable to mediate the effects of stress on the skin and may also influence the development of vitiligo [50].

Brain derived neurotrophic factor is a protein that belongs to the family of neurotrophin growth factors and is encoded by the BDNF gene. According to different studies, it has been found in the central and peripheral parts of the nervous system. This protein plays an important reinforcing role in the growth, differentiation and maintenance of neurons, allowing their longer survival [51]. Mental distress reduces BDNF level through activation of the hypothalamic-pituitary-adrenal axis. In addition, stress may cause the strain of sympathetic-adrenal-medullary axis, which increases the levels of cortisol and neuroinflammatory cytokines and reduces BDNF levels [52]. The role of neurotrophins and their receptors in the control of cutaneous hemostasis and hair growth was extensively studied [53]. Zhai et al. [54] report that neurotrophins mediate proliferative and survival signals in epidermal keratinocytes and, accordingly, influence proliferation and survival of melanocytes. Furthermore, some authors hypothesized that reduced serum BDNF levels may be an etiologic factor for the development of vitiligo and may also be an indicator of future psychiatric comorbidity in this category of patients [14].

This study has a number of limitations that have to be discussed. The major limitation comes from its nature of a cross-sectional study, so it was not possible to determine a cause-

and-effect relationship between the variables. Second, we utilized the ELISA kit that detected serum BDNF levels without identification of precursor and mature forms of this neuropeptide [14]. In addition, the sample size was rather modest. Still, we believe that our results might be confirmed on samples involving more cases and controls. Nevertheless, our study has several strengths. To our knowledge, this is the first investigation of *CRH-R1* rs242924 and *BDNF* rs11030094 polymorphisms in subjects with vitiligo. Also, we compared the serum levels of neurotransmitters with regard to the genotype subgroups of SNPs. At last, we explored a possible contribution of *CRH-R1* and *BDNF* gene variations to the pathogenesis of vitiligo. Our results indicated the association of both *CRH-R1* rs242924 and *BDNF* rs11030094 polymorphisms with the risk of vitiligo occurrence, but more research is certainly needed to confirm these results.

## Supporting information

**S1 Fig. Serum neurotransmitters levels (ng/ml) between controls and cases.**
(TIFF)

**S1 Table. Correlation analysis of BDNF, CRH levels and VES score.**
(TIFF)

**S1 Data.**
(XLSX)

## Acknowledgments

The authors of the article would like to express their gratitude to all patients who took part in the study.

## Author Contributions

**Conceptualization:** Nazira Bekenova, Almira Akhmetova, Yuliya Semenova.

**Data curation:** Almira Akhmetova, Almas Kussainov, Zhanar Urazalina, Oxana Yurkovskaya, Yerbol Smail, Laura Pak.

**Formal analysis:** Laura Kassym, Nazira Bekenova, Natalya Glushkova.

**Methodology:** Assiya Kussainova, Laura Kassym, Yuliya Semenova.

**Project administration:** Assiya Kussainova, Laura Kassym, Yuliya Semenova.

**Resources:** Almira Akhmetova, Almas Kussainov, Zhanar Urazalina, Oxana Yurkovskaya, Yerbol Smail, Laura Pak.

**Software:** Almas Kussainov, Zhanar Urazalina, Oxana Yurkovskaya, Laura Pak.

**Supervision:** Almira Akhmetova, Almas Kussainov, Zhanar Urazalina, Oxana Yurkovskaya, Yerbol Smail, Laura Pak, Yuliya Semenova.

**Validation:** Assiya Kussainova, Laura Kassym, Nazira Bekenova.

**Visualization:** Assiya Kussainova, Natalya Glushkova, Yerbol Smail.

**Writing – original draft:** Assiya Kussainova, Laura Kassym, Nazira Bekenova, Natalya Glushkova, Yuliya Semenova.

**Writing – review & editing:** Assiya Kussainova, Laura Kassym, Nazira Bekenova, Almira Akhmetova, Natalya Glushkova, Almas Kussainov, Zhanar Urazalina, Oxana Yurkovskaya, Yerbol Smail, Laura Pak, Yuliya Semenova.

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
