## [Decision Letter · Decision Letter 0]

26 Apr 2022

PONE-D-22-07470Gene polymorphisms and serum levels of BDNF and CRH in vitiligo patientsPLOS ONE

Dear Dr. Kassym,

Thank you for submitting your manuscript to PLOS ONE. After careful consideration, we feel that it has merit but does not fully meet PLOS ONE’s publication criteria as it currently stands. Therefore, we invite you to submit a revised version of the manuscript that addresses the points raised during the review process.

We look forward to receiving your revised manuscript.

Kind regards,

Livia D'Angelo

Academic Editor

PLOS ONE

Journal Requirements:

"This research was carried out as the part of PhD project"

"This research was carried out as the part of PhD project"

Reviewers' comments:

Reviewer's Responses to Questions

**Comments to the Author**

1. Is the manuscript technically sound, and do the data support the conclusions?

Reviewer #1: Partly

Reviewer #2: Yes

2. Has the statistical analysis been performed appropriately and rigorously? 

Reviewer #1: I Don't Know

Reviewer #2: No

3. Have the authors made all data underlying the findings in their manuscript fully available?

Reviewer #1: Yes

Reviewer #2: Yes

4. Is the manuscript presented in an intelligible fashion and written in standard English?

Reviewer #1: Yes

Reviewer #2: Yes

5. Review Comments to the Author

Reviewer #1: Assiya Kussainova and colleagues present an intersting question evalauting the serumm concentrations and polymorphisms in Brain-derived neurotrophic factor (BDNF) and Corticotropin releasing hormone (CRH) in relation to presence or absence of vitiligo.

Major comment

In the introduction psychological or psychiatric disorders are suggested to contribute to the etiology of vitiligo. I would suggest to the authors to rephrase this and stick to the concept psychological disorders may result from vitiligo in accordance with the references given. In the current phrasing the suggestion by the authors is not supported by the references given

The authors should add a brief discussion on the neuroedocine inflammatory axis iin the introduction to introduce their hypothesis. such as Sternberg EM. J Endocrinol. 2001

In the methods no infromation is provided on the psychological status of the vitiligo patients. In addition information on (auto)immune or inflammatory diseases in the controls is lacking. This should be provided or at least discussed.

The authro should discuss the effects of the polymorphisms studied on serum levels of the mediators as previosuly reported in literature.

Minor comments

fulll names of the factors measured should be given in the abstract, to make it comprehensible for readers not familiar with the abreviations.

Though written in comprehensible english, the manuscript would benefit if a native english speaker would edit the text.

The discussion should focus more on the potential effects from differences in BDNF and CRH expression levels on the neuroendocrine inflammatory axis, than on the association with neurological or psychiatric diseases.

Reviewer #2: 1. Please write the abstract starting with vitiligo and connect with BDNF and CRH, also please expand the full-form of BDNF and CRF.

2. Please rewrite the methodology in detail in the abstract.

3. Conclusion in the abstract is not reflecting the manuscript. Please rephrase it.

4. Please add the latest references in the introduction to describe the vitiligo (https://www.annsaudimed.net/doi/full/10.5144/0256-4947.2022.96). Please cite this reference as it is published in 2022.

5. Lane no 38/46/54/175/178/181/194/197/198/205/208/226, please update the reference as per the Plos One journal Style format. Replace the year and add the reference number.

6. Line no 70: Please add controls in the bracket after non-vitiligo

7. Please add the inclusion and exclusion criteria of cases and controls in detail

8. Please add the italics form for any name of the gene used in this study?

9. Lane no 107: What do you mean by 384 line number?

10. Hardy Weinberg Equilibrium, Authors can add the full form and add abbreviated form for using more than 1 time

11. Authors can add the genetic models in Both tables 2 and 4.

12. Elisa levels can be shown in the figure (Please use R/Python/Any other software)

13. Please add about BDNF and CRH and how it is correlated to Vitiligo?

14. Please add any meta-analysis studies if applicable?

15. Please focus on limitations of this study.

6. PLOS authors have the option to publish the peer review history of their article (what does this mean?). If published, this will include your full peer review and any attached files.

Reviewer #1: **Yes: **Michiel van der Flier

Reviewer #2: No

---

## [Author Response · Author response to Decision Letter 0]

17 May 2022

Journal Requirements:

Dear Editor, 

Thank you for considering our manuscript for publication in PLOS ONE. We did our best to respond to all comments made and to amend the manuscript accordingly.

We have removed the following information from the "Funding" section: This research was carried out as the part of PhD project and added: The author(s) received no specific funding for this work.

"This research was carried out as the part of PhD project"

"This research was carried out as the part of PhD project"

Kindly see the changes listed below to evaluate the work done by us.

Reviewer 1

Assiya Kussainova and colleagues present an intersting question evalauting the serumm concentrations and polymorphisms in Brain-derived neurotrophic factor (BDNF) and Corticotropin releasing hormone (CRH) in relation to presence or absence of vitiligo.

Thank you very much for reviewing our manuscript and for your thoughtful comments. Your suggestions are very valuable and helped us to improve the quality of our manuscript significantly.

We highlighted all changes made in yellow.

In the introduction psychological or psychiatric disorders are suggested to contribute to the etiology of vitiligo. I would suggest to the authors to rephrase this and stick to the concept psychological disorders may result from vitiligo in accordance with the references given. In the current phrasing the suggestion by the authors is not supported by the references given

Thank you. Done. As proposed, we introduced the following changes: 

Nevertheless, vitiligo may serve as a cause for is strongly associated with the wide range of psychological impairments resulting in violation of mental well-being and quality of life [4-6]

The authors should add a brief discussion on the neuroedocine inflammatory axis in the introduction to introduce their hypothesis. such as Sternberg EM. J Endocrinol. 2001

Done. We have added the following information to the "Introduction" section using recommended reference: Corticotropin-releasing hormone (CRH) regulates the hypothalamic-pituitary–adrenal axis (HPA) through activation of its receptor (corticotropin-releasing hormone receptor 1 (CRH-R1)). The CRH gene is also expressed in extracranial tissues, including human skin, and can be synthesized by a variety of immune cells. Human mast cells may contain sufficient amounts of CRH and other stress peptides to mediate its autoimmune and anti-inflammatory effects [16]. Expression of CRH and CRH-R1 mRNA also has been detected in cultured human melanocytes and melanoma cells [17]. Research on vitiligo demonstrated that expression of the CRH-R1 gene is increased in both vitiligo unaffected and vitiligo affected skin. Furthermore, the increased expression of CRH and CRH-R1 correlated with high scores obtained on the Social adaptation assessment scale, which can be explained by the fact that hyperactivation of the cortical-releasing system in skin increases neuroendocrine sensitivity to stress [18].

In the methods no information is provided on the psychological status of the vitiligo patients. 

Done. We have entered the following information in the "Materials and Methods” section: 

The psychological status of study participants was identified using Generalized Anxiety Disorder-7 (GAD-7) scale. The GAD-7 scores range from 0 to 21, with 0-4 none, 5-9 mild, 10-14 moderate and 15 and more severe levels of anxiety symptoms [25].

In addition information on (auto)immune or inflammatory diseases in the controls is lacking. This should be provided or at least discussed.

Thank you. To overcome this drawback, we added this information to the “Study population section”: 

Patients were ineligible for this investigation if they were (i) physically or mentally unable to give informed consent, (ii) had a history of any psychiatric disorder, (iii) had some somatic conditions, such as heart, pulmonary diseases or diabetes mellitus that affected their mental status, (iv) had other skin problems (acne, psoriasis, eczema, etc.)., and (v) presented with autoimmune or inflammatory diseases.

The authro should discuss the effects of the polymorphisms studied on serum levels of the mediators as previosuly reported in literature.

Done. We added the following piece of information to the “Discussion” section: 

Unfortunately, there is a lack of evidence reporting the effects of the polymorphisms studied on serum levels of the mediators. Two SNP interactions (rs242941 and rs1876828 of the CRH-R1 gene) were linked to the serum cortisol levels [32].

 There is no data on associations of SNPs with serum levels of BDNF in patients with the skin disorders. Few studies described such associations between SNPs of other mediators and their blood levels. For instance, a significant association between IL-6 572 gene polymorphism and its serum level was reported in Egyptian patients with keloid [36]. Another study showed that TT genotype of IL-19 might be a hereditary risk factor for acne vulgaris development, and it was associated with elevated IL-19 serum levels, which could be a marker of acne severity [37].

fulll names of the factors measured should be given in the abstract, to make it comprehensible for readers not familiar with the abreviations

Thank you. Done. 

The genes encrypting brain-derived neurotrophic factor (BDNF) and corticotropin releasing hormone (CRH) might be the conceivable contributors to the development of vitiligo. …… Genotyping for the rs11030094 polymorphism of the BDNF gene and rs242924 of the corticotropin releasing hormone receptor 1 (CRH-R1) gene was performed by real-time polymerase chain reaction (PCR) ……

Though written in comprehensible english, the manuscript would benefit if a native english speaker would edit the text.

Done. We didn’t highlight the areas of improvement as in this case the entire manuscript would be highlighted.

The discussion should focus more on the potential effects from differences in BDNF and CRH expression levels on the neuroendocrine inflammatory axis, than on the association with neurological or psychiatric diseases.

We addressed the issue of information overload related to the link between neurotransmitters and mental disorders and added the following pieces of information to the “Discussion” section: The variations of BDNF BDNF gene were described in several studies related to the impairment of the essential brain functions. Honea et al. (2013) [28] revealed that BDNF BDNF rs11030094 polymorphism was associated with cognitive decline in the patients with Alzheimer’s disease. Warburton et al. (2016) [17] linked BDNF BDNF rs11030094 SNP to memory dysfunction in patients with newly diagnosed epilepsy. The recent study of Hennings et al. (2019) [20] reported on the association between BDNF BDNF rs11030094 polymorphism and antidepressant response in subjects with major depressive disorder.

 Our recent study also demonstrated the decreased level of serum BDNF and increased level of serum CRH in vitiligo patients in comparison with the healthy controls [33].

 BDNF signaling may be engaged in affective behavior in the form of environmental stresses that cause depression, as well as reduce BDNF mRNA [42]. Based on this, BDNF is extensively investigated in neuropsychiatric diseases such as schizophrenia, severe depressive disorder, MDD, bipolar disorder and various addictions [43]. In the majority of these studies psychiatric disorders were found to be associated with decreased BDNF levels. Moreover, other reports have demonstrated that adequate efficient therapy improves BDNF levels in these patients [44].

Mental distress reduces BDNF level through activation of the hypothalamic-pituitary-adrenal axis. In addition, stress may cause the strain of sympathetic-adrenal-medullary axis, which increases the levels of cortisol and neuroinflammatory cytokines and reduces BDNF levels [52].

Reviewer 2

Thank you very much for reviewing our manuscript. Your comments helped us to improve its quality.

We highlighted all changes made in yellow

1. Please write the abstract starting with vitiligo and connect with BDNF and CRH, also please expand the full-form of BDNF and CRF.

Done. As proposed, we introduced the following changes: Vitiligo is one of the most common hypomelanoses, in which the destruction of functioning melanocytes causes depigmentation of the skin, hair and mucous membranes. 

The genes encrypting brain-derived neurotrophic factor (BDNF) and corticotropin releasing hormone (CRH) might be the conceivable contributors to the development of vitiligo.

2. Please rewrite the methodology in detail in the abstract.

Done:

The cross-sectional study was carried out between October 2020 and June 2021 in 93 vitiligo patients (age range from 23 to 48 years) and 132 healthy controls (age range from 24 to 52 years). The psychological status of study participants was evaluated using the Generalized Anxiety Disorder-7 (GAD-7) scale. Serum levels of BDNF and CRH were measured with the help of a commercially available sandwich enzyme-linked immunosorbent assay (ELISA) kit. Genotyping for the rs11030094 polymorphism of the BDNF gene and for the rs242924 polymorphism of the corticotropin releasing hormone receptor 1 (CRH-R1) gene was performed by a real-time polymerase chain reaction (PCR).

3. Conclusion in the abstract is not reflecting the manuscript. Please rephrase it.

Thank you. We rewrote the conclusion as recommended:

 These findings may point to a potential role for the aforementioned genetic variations in the development of vitiligo. 

 Our findings demonstrated the association between CRH-R1 rs242924 and BDNF rs11030094 polymorphisms and vitiligo. Further studies need to be carried out in vitiligo patients to confirm the results observed.

4. Please add the latest references in the introduction to describe the vitiligo (https://www.annsaudimed.net/doi/full/10.5144/0256-4947.2022.96).

Thank you. We have removed previous information from the text and replaced it by the following statement: Vitiligo is a chronic autoimmune skin disease affecting 0.5%-2% of population worldwide [1]. 

Vitiligo is an acquired pigmentry disorder of the skin and mucous membranes which manifests as white macules due to selective loss of melanocytes [1].

Please cite this reference as it is published in 2022.

Done. 1. Bergqvist C, Ezzedine K. Vitiligo: A Review. Dermatology. 2020;236(6):571-592. doi: 10.1159/000506103. Epub 2020 Mar 10. PMID: 32155629. 

1. Saif GB, Khan IA. Association of genetic variants of the vitamin D receptor gene with vitiligo in a tertiary care center in a Saudi population: a case-control study. Ann Saudi Med. 2022 Mar-Apr;42(2):96-106. doi: 10.5144/0256-4947.2022.96. Epub 2022 Apr 7. PMID: 35380061; PMCID: PMC8982003.

5. Lines no 38/46/54/175/178/181/194/197/198/205/208/226, please update the reference as per the Plos One journal Style format. Replace the year and add the reference number.

Done.

6. Line no 70: Please add controls in the bracket after non-vitiligo

Done: 

The non-vitiligo (control) participants were matched by age and sex as the comparator group, and were recruited through community advertisement.

7. Please add the inclusion and exclusion criteria of cases and controls in detail

Thank you. Done: 

For the patient to be considered eligible to take part in the study, the following inclusion criteria had to be met: (i) age of 16 years and older, (ii) a verified diagnosis of vitiligo, and (iii) an ability to read and understand Russian or Kazakh. Patients were ineligible for this investigation if they were (i) physically or mentally unable to give informed consent, (ii) had a history of any psychiatric disorder, (iii) had some somatic conditions, such as heart, pulmonary diseases or diabetes mellitus that affected their mental status, (iv) had other skin problems (acne, psoriasis, eczema, etc.)., and (v) presented with autoimmune or inflammatory diseases.

8. Please add the italics form for any name of the gene used in this study?

Thank you very much. Done.

9. Line no 107: What do you mean by 384 line number?

Thank you. We have removed this information from the text: “….and 10 ng of DNA in a total volume of 5 µl. in 384-well plates (Roche, Indianapolis, IN, USA).”

10. Hardy Weinberg Equilibrium, Authors can add the full form and add abbreviated form for using more than 1 time.

Done:

 Both case and control groups were in Hardi-Weinberg equilibrium HWE.

11. Authors can add the genetic models in Both tables 2 and 4.

Thank you. Done.

Table 2. Genotype distribution and allele frequencies of CRH-R1 CRH-R1 rs242924 

Alleles Cases

N = 93 (%) Controls

N = 132 (%) P OR (95% CI)

G 24 (25.3%) 52 (39.4%) 0.002 0.52 (0.34-0.79)

T 69 (74.7%) 80 (60.6%) 1.92 (1.27-2.90)

Genotypes

GG 6 (6.5%) 28 (21.2%) 

0.007 0.26 (0.10-0.65)

GT 35 (37.6%) 48 (36.4%) 1.06 (0.61-1.83)

TT 52 (55.9%) 56 (42.4%) 1.72 (1.01-2.94)

P < 0.05 is significant 

OR – Odds Ratio (95% CI - 95% Confidence Interval)

A multiplicative model (df=1) was used to identify the association of alleles with the disease

A general inheritance model (df=2) was used to identify the association of genotypes with the disease

Table 4. Genotype distribution and allele frequencies of BDNF BDNF rs11030094

Alleles Case Control р OR (95% CI)

G 53 (57.5%) 61 (46.2%) 0.002 1.58 (1.08-2.30)

A 40 (42.5%) 71 (53.8%) 0.63 (0.43-0.93)

Genotypes

GG 28 (30.1%) 33 (25.0%) 

0.01 1.29 (0.71-2.34)

GA 51 (54.8%) 56 (42.4%) 1.65(0.97-2.81)

AA 14 (15.1%) 43 (32.6%) 0.37(0.19-0.72)

 P < 0.05 is significant 

OR – Odds Ratio (95% CI - 95% Confidence Interval)

A multiplicative model (df=1) was used to identify the association of alleles with the disease

A general inheritance model (df=2) was used to identify the association of genotypes with the disease

12. Elisa levels can be shown in the figure (Please use R/Python/Any other software)

We have added this information to the Supplement materials: 

The serum levels of BDNF and CRH are presented on S1 Fig.

13. Please add about BDNF and CRH and how it is correlated to Vitiligo?

Done. We have added this information to the “Materials and methods” section: The percentage of skin lesions was calculated using the Vitiligo Extent Score (VES) online calculator [24]. 

The correlation of ELISA levels and VES was not significant (S1 Table). – “Result” section

to the Supplement materials:

S1 Table. Correlation analysis of BDNF, CRH levels and VES score

 BDNF CRH

VES score r -.031 -.005

 p-value .769 .961

14. Please add any meta-analysis studies if applicable?

Thank you. We have added the following information to the 

"Discussion" section: 

Nevertheless, some evidence on association of vitiligo with gene polymorphisms exists. Tumor necrosis factor (TNF)-α- 308 G/A gene polymorphism was the most investigated in vitiligo patients. The recent meta-analysis of Giri et al. [38] suggested the involvement of susceptible 'A' allele with vitiligo susceptibility in overall population, whereas the previous studies failed to reveal that TNF-α-308 G/A polymorphism is a genetic risk factor for vitiligo [39, 40].

15. Please focus on limitations of this study.

Thank you. We have added the following information to the 

"Discussion" section: 

This study has a number of limitations that have to be discussed. The major limitation comes from its nature of a cross-sectional study, so it was not possible to determine a cause-and-effect relationship between the variables. Second, we utilized the ELISA kit that detected serum BDNF levels without identification of precursor and mature forms of this neuropeptide [14]. In addition, the sample size was rather modest. Still, we believe that our results might be confirmed on samples involving more cases and controls.

---

## [Decision Letter · Decision Letter 1]

7 Jul 2022

Gene polymorphisms and serum levels of BDNF and CRH in vitiligo patients

PONE-D-22-07470R1

Dear Dr. Kassym,

We’re pleased to inform you that your manuscript has been judged scientifically suitable for publication and will be formally accepted for publication once it meets all outstanding technical requirements.

Kind regards,

Livia D'Angelo

Academic Editor

PLOS ONE

Additional Editor Comments (optional):

Reviewers' comments:

Reviewer's Responses to Questions

**Comments to the Author**

1. If the authors have adequately addressed your comments raised in a previous round of review and you feel that this manuscript is now acceptable for publication, you may indicate that here to bypass the “Comments to the Author” section, enter your conflict of interest statement in the “Confidential to Editor” section, and submit your "Accept" recommendation.

Reviewer #2: All comments have been addressed

2. Is the manuscript technically sound, and do the data support the conclusions?

Reviewer #2: Yes

3. Has the statistical analysis been performed appropriately and rigorously? 

Reviewer #2: Yes

4. Have the authors made all data underlying the findings in their manuscript fully available?

Reviewer #2: Yes

5. Is the manuscript presented in an intelligible fashion and written in standard English?

Reviewer #2: Yes

6. Review Comments to the Author

Reviewer #2: The authors have worked very hard to justified all the raised comments. Therefore, the manuscript can be accepted in the current form.

7. PLOS authors have the option to publish the peer review history of their article (what does this mean?). If published, this will include your full peer review and any attached files.

Reviewer #2: **Yes: **Dr. Imran Ali Khan

---

## [Editor Report · Acceptance letter]

21 Jul 2022

PONE-D-22-07470R1 

Gene polymorphisms and serum levels of BDNF and CRH in vitiligo patients 

Dear Dr. Kassym:

I'm pleased to inform you that your manuscript has been deemed suitable for publication in PLOS ONE. Congratulations! Your manuscript is now with our production department. 

Kind regards, 

on behalf of

Dr. Livia D'Angelo 

Academic Editor

PLOS ONE